# Chronic Autoimmune Gastritis: Modern Diagnostic Principles

**DOI:** 10.3390/diagnostics11112113

**Published:** 2021-11-15

**Authors:** Maria A. Livzan, Olga V. Gaus, Sergei I. Mozgovoi, Dmitry S. Bordin

**Affiliations:** 1Omsk Sate Medical University, 644099 Omsk, Russia; mlivzan@yandex.ru (M.A.L.); gaus_olga@bk.ru (O.V.G.); simozgovoy@yandex.com (S.I.M.); 2A.S. Loginov Moscow Clinical Scientific Center, 111123 Moscow, Russia; 3A.I. Yevdokimov Moscow State University of Medicine and Dentistry, 127473 Moscow, Russia; 4Tver State Medical University, 170100 Tver, Russia

**Keywords:** chronic autoimmune gastritis, atrophic gastritis, nutritional status, vitamin B12 deficiency, megaloblastic anemia, neuroendocrine tumors, gastric adenocarcinoma

## Abstract

This article summarizes and systematizes the available data from the literature on chronic autoimmune gastritis (CAG) in order to increase the awareness of specialists about the modern possibilities for diagnosing the disease, including its early stages. The clinical manifestation of the disease includes possible variants such as gastrointestinal, hematological (first of all, the formation of iron deficiency and B12-deficiency anemia), and neurological variants. Patients with chronic autoimmune gastritis are characterized by comorbidity with other autoimmune diseases. In this paper, data on the most informative serological markers for the diagnosis of CAG, as well as laboratory tests to detect micronutrient deficiencies, information on the characteristic changes in the gastric mucosa, and the prognosis of the disease, are presented. The diagnosis of CAG should be based on a multidisciplinary approach that combines a thorough analysis of a patient’s complaints with a mandatory assessment of nutritional status, as well as the results of serological, endoscopic, and histological research methods.

## 1. Introduction

Helicobacter pylori (HP) infection and the autoimmune inflammation of the gastric mucosa are recognized as the leading etiological factors of chronic atrophic gastritis, a disease associated with an increased risk of stomach cancer development. Additionally, in relation to HP-associated gastritis, there is international and national consensus on the diagnosis and treatment of infection [1,2]; however, with regard to autoimmune inflammation, clinicians have faced diagnostic difficulties. At the same time, autoimmune inflammation of the gastric mucosa, in addition to the formation of atrophy and an increase in the risk of stomach cancer, carries additional risks, both in relation to neoplasms (neuroendocrine tumors) and in relation to the deficiency of a number of micronutrients with the involvement of other organs and systems.

The results of individual epidemiological studies appearing in the literature [3,4] indicate that clinicians underestimate the abovementioned variant for the development of atrophy, which requires joint efforts aimed at increasing the awareness of specialists about modern diagnostic capabilities.

This article aims at systematizing the available data on approaches to the diagnosis of chronic autoimmune gastritis (CAG), an immune-mediated inflammatory disease of the gastric mucosa associated with the synthesis of autoantibodies to parietal cells and intrinsic Castle’s factor [5].

Paradoxically, the studies on CAG began long before the first ideas about the existence of the disease itself. In 1849, Thomas Addison, and then in 1872, Anton Biermer, described a so-called “idiopathic anemia” that led to the rapid death of patients and further developed the name “pernicious anemia” (from Lat. *perniciosus*—fatal, dangerous). In 1926, a group of American scientists, including George Hoyt Whipple, George Richards Minot, and William Parry Murphy, experimentally established that pernicious anemia could be treated by introducing raw liver into the diet and that the disease was based on the inability of the stomach glands to secrete a substance necessary for absorption by the intestine of a water-soluble extract isolated from the liver. In 1934, these scientists were awarded the Nobel Prize in Physiology or Medicine for their discovery. However, the isolation of a separate form of gastritis, leading to the development of B12-deficiency anemia, type A (autoimmune) gastritis, only took place in 1973, which was possible due to the identification of autoantibodies to autoantigens, particularly antibodies to parietal cells, α- and β- subunits of H +/K + -ATPase in the stomach.

### 1.1. Risk Factors

As with other autoimmune diseases, the etiological factor has not been established for CAG; today, we can only talk about risk factors for the disease, such as genetic and environmental factors. It has been reported that the risk of CAG development is associated with the HLA-D8 and HLA-DR-3 haplotypes of the major histocompatibility system [6]. The disease is more common in women (female-to-male ratio is 3:1) and in individuals aged over 60, although a recently published Swedish prospective study showed a higher incidence in the 35 to 45 age group [7]. As such, the previously assumed link between CAG and European heritage is now in doubt [6].

The role of infectious agents as triggering factors of the disease has been discussed elsewhere. In particular, it has been shown that the detection rate of CAG is statistically significantly higher among children infected with the Epstein–Barr virus [8]. In addition, in the available modern literature, the role of HP in the development of CAG is highly debatable. Some authors, relying on the hypothesis of the molecular mimicry existence between the HP antigens and the H +/K + -ATPase of parietal cells, insist that infection is a disease trigger in genetically predisposed individuals; others, referring to the heterogeneity of such data in epidemiological studies, refute this link [9,10,11,12].

The role of nutrition in the development of autoimmune inflammation has not been reliably established. It is believed that a widespread Western diet, which is high in calories, simple carbohydrates, refined foods, saturated fats, and red meat, and low in omega-3 and omega-9 polyunsaturated fatty acids and plant fiber, may increase the risk of autoimmune diseases [13,14]. For the prevention and inhibition of autoimmune inflammation, an autoimmune protocol diet was developed, which is a modification of the Paleolithic diet with the elimination of specific foods, including cereals, beans, solanaceous family, dairy products, eggs, coffee, alcohol, nuts and seeds, refined sugars, oils, and food additives [15].

### 1.2. Clinical Manifestation

CAG can be asymptomatic or can be characterized by the presence of nonspecific gastrointestinal symptoms, which is why the disease may remain unrecognized for a long time [16]. As the disease progresses and the atrophy of the gastric mucosa develops, various hematological and neuropsychiatric disorders occur. It is reported that, at the initial diagnosis, hematological manifestations occur in 37% of patients, and neurological ones occur in less than 10% of cases [17]. The list of possible clinical manifestations of CAG is presented in Table 1.

### 1.3. Gastrointestinal Symptoms

Traditionally, “chronic gastritis”, regardless of its etiology, is considered a morphological diagnosis and has no clinical equivalents. However, upon questioning, certain gastrointestinal symptoms can be detected in most patients. It has been shown that more than 55% of patients with CAG have concomitant manifestations of functional dyspepsia in the form of postprandial distress syndrome [18]. In their work, Kalkan et al., using the method of scintigraphy, demonstrated that, in 80% of cases, dyspepsia syndrome in CAG was associated with delayed gastric emptying, probably due to changes in the motility of the antrum in conditions of hypergastrinemia [19]. Heartburn and regurgitation are present in 24% and 12% of patients with CAG, respectively [20]. At the same time, according to 24 h multichannel pH impedance monitoring, non-acid refluxes are more common among individuals with CAG; therefore, the use of antisecretory drugs, as a rule, is unjustified and clinically ineffective [21].

CAG is characterized by a decrease in the absorption of valuable micronutrients, primarily vitamin B12 and iron, with the development of their deficiency symptoms, which can help in identifying such patients (Table 2).

### 1.4. Vitamin B12 (Cobalamin) Deficiency

Vitamin B12 is absorbed in the distal parts of the small intestine after it binds to the intrinsic Castle’s factor synthesized by the parietal cells of the stomach. Cobalamin, the only one of all water-soluble vitamins, forms a depot in the liver, the reserves of which are sufficient for 4–5 years, even after the complete cessation of vitamin intake. As such, B12’s long deficiency can be sufficiently compensated with no clinical manifestation.

The biological role of vitamin B12 is extremely diverse. It acts as a co-factor in various metabolic processes, including the reaction of the homocysteine conversion into methionine, which underlies the synthesis of purine bases necessary for DNA building. This explains the negative effect of vitamin B12 deficiency on hematopoiesis, with the disorder being characterized by the production of erythroid lineage cells in the bone marrow or their premature destruction (ineffective erythropoiesis) and the development of megaloblastic anemia.

Typical manifestations of anemic syndrome are weakness, fatigue, dizziness, hypotension and compensatory tachycardia, and shortness of breath during or following exertion. Pallor of the skin with B12-deficiency anemia is often combined with lemon yellowness due to intramedullary hemolysis and, as a consequence, an increase in the level of bilirubin in the blood serum.

Another consequence of vitamin B12 deficiency is neuropathy, because B12 vitamin is necessary for the synthesis of the myelin sheaths of nerve endings, which ensure the conduction of nerve impulses. It should be noted that neurological disorders can be detected even in the absence of hematological changes [22]. Specific damage to the nervous system with a deficiency of vitamin B12 is called “funicular myelosis”. The most common sensory–motor peripheral polyneuropathies (25% of cases) are numbness in the hands and feet (“glove-and-stocking syndrome”), a tingling sensation in the distal parts of the hands and feet, numbness, cold, a feeling of swelling, or compression from the outside. Less often (1–2%), spastic paraparesis, sensory ataxia, visual or hearing impairments, unsteady gait, urinary disorders, and changes in tendon and extrapyramidal reflexes are observed. Cognitive impairment, apathy, and depression can also occur [16].

Pathognomonic for B12-deficiency anemia is the development of Hunter atrophic glossitis with the presence of a crimson lacquered tongue. Often, patients complain of a burning sensation at the tip of the tongue and a feeling that the tongue does not fit in the oral cavity. Objectively, attention is drawn to the bright crimson color, the smoothness of the papillae, and the imprints of the teeth on the lateral surfaces of the tongue.

Vitamin B12 deficiency, even at early stages, sometimes before the development of clinical manifestations, leads to the homocysteine concentration increase in serum, which is proportional to the severity of the B12 deficiency [23]. Homocysteine is a sulfur-containing amino acid that is synthesized from methionine in the remethylation cycle, which also requires the presence of vitamin B12 and folic acid as co-factors. Elevated homocysteine concentrations in plasma are recognized as independent risk factors for cardiovascular disease and also appear to play an important role in the development of dementia, diabetes mellitus, and renal failure. The direct toxic effect of homocysteine on endothelial cells and impairment of endothelium-dependent vasodilation under conditions of hyperhomocysteinemia, leading to progressive damage to the intima of the vascular wall, have been proven [16]. The clinical significance of hyperhomocysteinemia and the frequency of its association with pathology of the cardiovascular system in patients with CAG remains to be assessed.

### 1.5. Iron Deficiency

IDA is detected in 52% of patients with CAG. In 35–58% of cases, it precedes the onset of B12-deficiency anemia and is currently considered the most frequent hematological manifestation of the disease [24,25]. The pathophysiology of iron deficiency can apparently be associated with several sources, such as the presence of erosive damage to the mucous membrane and possible latent blood loss, concomitant HP infection and bacteria competition for dietary iron, hypochlorhydria, and increased hepcidin synthesis against the background of an existing inflammatory process of any localization.

### 1.6. Concomitant Diseases (Comorbidity)

According to epidemiological studies, the prevalence of concomitant autoimmune diseases in patients with CAG reaches 40%, and the most common disorders are thyroid diseases, type 1 diabetes mellitus, hemolytic anemia, vitiligo, alopecia, rheumatoid arthritis, psoriasis, autoimmune hepatitis, myasthenia, and Sjogren disease [18]. The spectrum of autoimmune diseases associated with CAG is presented in Table 3.

## 2. Diagnostic Approaches

There is no gold standard for the diagnosis of CAG, so the diagnosis is established on the basis of a set of data from a survey and examination of a patient, as well as the results of laboratory and instrumental research methods.

### 2.1. Laboratory Methods

Serological research methods are not an independent diagnostic tool, since they do not have absolute specificity and sensitivity in relation to CAG. Thus, antibodies to parietal cells are considered a highly sensitive marker and are found in 80–90% of patients with CAG, but they can also be present in 7.8–19.5% of healthy adults, as well as in persons infected with HP or with other autoimmune diseases such as type 1 diabetes mellitus or Hashimoto’s thyroiditis [26,27]. The determination of antibodies to parietal cells using the method of immunofluorescence or ELISA increases the sensitivity of the test by 30%. It should be noted that, in the later stages of the disease, as the atrophy of the gastric mucosa progresses, the loss of the target autoantigen antibodies to parietal cells may not be detected [26]. Antibodies to the intrinsic Castle’s factor are more specific for CAG (98.6%), but their sensitivity is not high (30–60%) [28,29]. At the same time, it was found that antibodies to the intrinsic Castle’s factor correlated well with atrophy of the gastric mucosa; this was revealed by histological examination of gastrobiopsy specimens [30].

Atrophy and loss of functional activity of parietal cells of the gastric body with the development of hypo- or achlorhydria, on the one hand, leads to hyperplasia of gastrin-producing cells of the antrum and hypergastrinemia, and on the other hand, to a decrease in the production of pepsinogen I. At the same time, if the pathological process does not involve the glands of the antrum, the level of pepsinogen II does not change significantly, but the ratio of pepsinogen I to pepsinogen II decreases, which, together with hypergastrinemia, are characteristic signs of CAG [26]. These indicators are included in the test “GastroPanel” (Biohit, Finland), which received the figurative name “serological biopsy” [31,32]. It has been reported that a decrease in pepsinogen I of less than 30 μg/L and a pepsinogen I/pepsinogen II ratio of less than three, in combination with an increase in gastrin-17 levels of more than 30 pmol/L, reliably helps in the identification of asymptomatic patients with CAG at the stage of atrophy [33]. In this case, the sensitivity of the method is 74.5% and the specificity reaches 100%.

It is fundamentally important for both diagnosis and prognosis to assess a patient’s nutritional status with a mandatory study of cyanocobalamin, iron, ferritin, total iron-binding capacity, calcium, vitamin D, and ascorbic acid levels in the blood serum.

Determination of chromogranin A in blood serum, whose level correlates with the degree of the ECL cells’ hyperplasia, can be considered as an auxiliary marker for assessing the risk of NET formation [34,35,36,37]. The sensitivity and specificity of chromogranin A as a possible marker of NET is low; therefore, its determination result should be assessed taking into account the clinical picture and the histological examination data. For example, the concentration of chromogranin A can increase in patients with inflammatory bowel disease, hepatocellular carcinoma, non-alcoholic fatty liver disease, renal failure, and long-term use of proton pump inhibitors [38,39]. The summarized data on laboratory markers of CAG are presented in Table 4.

### 2.2. Endoscopic Examination

Endoscopic examination plays an important role in the diagnosis of CAG, since the quality of the morphological assessment of gastric mucosa damage depends on the adequacy of the endoscopist’s actions while collecting gastrobiopsy specimens. At the initial stages of CAG, the endoscopic picture may not be changed or correspond to minimal inflammatory changes, which are more pronounced in the body of the stomach. Macroscopically, the gastric mucosa in patients with CAG is usually thinner than normal, with a smoothing of the relief and disappearance of folds, as well as the presence of highly visible vessels of the stomach’s submucosa. As the atrophy of the glands progresses, an unevenly expressed flattening of the relief of the mucous membrane occurs in the area of the stomach’s body and its fundus. Uneven distribution of the atrophy’s foci gives the surface of the mucous membrane a pseudopolypiform character due to the presence of intact areas [3]. Such a macroscopic picture forces one to take biopsy material from different parts of the gastric mucosa during endoscopic examination, given the flattened and polypoid nature of the surface, which makes it possible to adequately assess the nature of the changes.

The diagnostic value of endoscopic signs of CAG depends on the research method utilized.

According to the Guidelines for the Management of Precancerous Conditions and Lesions in the Stomach (MAPS II), in the diagnosis and staging of precancerous conditions and changes in the gastric mucosa, the capabilities of high-resolution endoscopy in combination with chromoscopy are higher than those of white light endoscopy [40]. Whenever possible, virtual chromoscopy should be used during biopsy for staging the atrophic process, as well as to identify suspicious areas of neoplasia and conduct targeted biopsy. For an adequate assessment of precancerous diseases of the stomach during endoscopic examination, it is necessary to perform a biopsy from at least four points: from the stomach’s antrum and body, along the lesser and greater curvature. If the risk of gastric cancer is assessed using the Operative Link for Gastritis Assessment of Atrophic Gastritis (OLGA) or Operative Link for Gastric Intestinal Metaplasia (OLGIM) systems, an additional biopsy sample from the corner of the stomach is advised. The recommended frequency of endoscopic examination in patients with autoimmune gastritis is once every 3–5 years.

### 2.3. Morphological Examination

Histological examination of gastrobiopsy specimens is the gold standard for diagnosing chronic gastritis as well as for assessing the presence and severity of inflammatory or atrophic changes.

In microscopic examination, the main feature of the uncomplicated course of CAG (that is to say, in the absence of dysplasia, cancer, or neuroendocrine hyperplasia) is a diffuse lesion of the mucous membrane, limited to the area of the stomach body [41]. The state of the mucous membrane of the stomach’s antrum may correspond to a normal histological structure or may have a mononuclear inflammatory infiltrate, focal absolute, and/or metaplastic atrophy. Many patients may also show signs of foveolar hyperplasia (“reactive gastropathy”), probably reflecting the effect of hypergastrinemia, with a decrease in mucus-producing activity of the integumentary epithelium. As a rule, hyperplasia of gastrin-producing cells is present. In the case of the addition of HP infection (combined lesion), all the corresponding signs of damage to the mucous membrane of the antrum appear.

In the mucous membrane of the stomach’s body, a wide range of atrophic changes (from minimal signs of inflammation to severe atrophy of the glands) can be detected, characterized by varying degrees of the acid-producing glands’ damage, as well as the development of intestinal, pyloric (pseudopyloric metaplasia), and ECL cell hyperplasia. The process of restructuring the mucous membrane can be divided into three stages or phases. In the first, initial stage, there is a diffuse or multifocal infiltration of the lamina propria of the mucous membrane by lymphocytes and plasma cells up to the formation of pronounced infiltration and involvement of the entire thickness of the mucous membrane, often with an admixture of eosinophils and mast cells (Figure 1). Focal destruction of individual glands by components of the inflammatory infiltrate takes place. Hyperplasia of parietal cells is also characteristic, reflecting the state of hypergastrinemia. At this stage of the disease, intestinal metaplasia is rare and focal [42,43].

In the second stage, or expanded (florid) stage, pronounced atrophy of acid-producing glands develops; diffuse lymphoplasmacytic infiltration of the lamina propria is characteristic of a decrease in the height of the glands and an increase in the thickness of the foveolar component. The prevalence of intestinal metaplasia is higher than in the previous phase, but pseudopyloric metaplasia plays the main role in the remodeling of gastric mucosa (Figure 2). In recent years, this type of metaplasia, with the formation of glands resembling the antral glands, is associated with a separate phenomenon—a cell line with the expression of spasmolytic polypeptide (SPEM—spasmolytic polypeptide-expressing metaplasia). The appearance of this cellular rearrangement is associated with an increased risk of an intestinal-type gastric cancer developing in patients with CAG [44,45,46]. In this stage, due to the presence of pronounced atrophic changes with selective involvement of the stomach’s glands, the morphological picture becomes pathognomonic. However, establishing the presence of antibodies to parietal cells and intrinsic factors also remains necessary to confirm the diagnosis.

In the final stage of the disease, there is a pronounced reduction in acid-producing glands with the development of foveolar hyperplasia, the appearance of hypermucoidized cells, the formation of hyperplastic polyps, and an increase of the areas with pyloric (pseudopyloric), pancreatic, and intestinal metaplasia (Figure 3).

The classification of intestinal metaplasia is based on histochemical evaluation of mucins, as follows [47,48]: Complete intestinal metaplasia (type I): non-secretory absorptive cells with brush border and sialomucin-secreting goblet cells. Incomplete intestinal metaplasia type II with columnar cells secreting neutral and acid sialomucin, and goblet cells secreting mainly sialomucin but occasionally sulphomucin. Incomplete intestinal metaplasia type III with columnar cells secreting predominantly sulphomucin and goblet cells secreting sialomucin or sulphomucin.

The thickness of the muscle plate of the mucous membrane can be increased by three to four times. At this stage, parietal cells are difficult to detect, inflammatory infiltration is minimal or absent, and individual clusters of lymphoid cells may remain [43].

To stratify the risk of stomach cancer development in a patient with chronic gastritis, a system of morphological assessment to detect changes in the gastric mucosa was proposed—the Operative Link for Gastritis Assessment of Atrophic Gastritis (OLGA). Integral indicators of the degree and stage of chronic gastritis are determined according to this system, where the degree is understood as the infiltration severity of the lamina propria of the gastric mucosa by inflammatory cells (lymphocytes, plasma cells, and neutrophilic leukocytes). The stage is understood as the presence of atrophic changes, including intestinal metaplasia [49]. It should be emphasized that, with the increase in the stage of chronic gastritis, the likelihood of developing adenocarcinoma of the stomach increases. For example, in patients with stage III–IV, the risk increases by five to six times. In addition, an increased risk of gastric cancer is noted among individuals with incomplete (colonic, type III) and/or widespread intestinal metaplasia of the mucous membrane [50,51]. The modified OLGIM system (Operative Link on Gastric Intestinal Metaplasia Assessment), being more reproducible, offers only an assessment of the intestinal metaplasia presence as the main indicator of chronic gastritis stage [52,53]. Although the question of comparing these classifications from the sensitivity point remains controversial, since the use of OLGIM, which is more reproducible in practical application, may underestimate the true severity of absolute (non-metaplastic) atrophy [54,55,56].

During the extended and final stage of the disease, due to hypergastrinemia, the proliferation of neuroendocrine ECL cells of the gastric body is stimulated. A similar process also occurs in Zollinger–Ellison syndrome, multiple endocrine neoplasia syndromes, and HP-associated multifocal atrophic gastritis.

The following forms of hyperplasia of neuroendocrine cells in the gastric mucosa are distinguished [57,58]:Simple diffuse hyperplasia. This is characterized by a more than two-fold increase in the population of ECL cells. Diagnosis is difficult due to the lack of clear quantitative criteria. The diagnosis is poorly reproduced on biopsy material.Linear hyperplasia. The presence in one visual field of at least two groups of linearly located neuroendocrine cells, consisting of five or more cells. Usually, changes are diagnosed in the area of the neck glands (Figure 4).Micronodular hyperplasia. The presence of the cells’ cluster in contact with the basement membrane, but not exceeding the diameter of the gland, up to 150 μm in diameter, or a similar cluster located freely in the lamina propria of the mucous membrane.Adenomatous (adenomatoid) hyperplasia. The presence of an aggregate of five or more clusters (Figure 5).Neuroendocrine cells dysplasia. Merging clusters with diameters of more than 150 µm but less than 500 µm.

NETs associated with ECL cell hyperplasia occur in 5–8% of patients with autoimmune gastritis and comprise 70 to 80% of all gastric NETs [59]. Typically, these tumors have a good prognosis, with a five-year survival in more than 95% of patients. This is significantly different from the prognosis of sporadic tumors, which are more aggressive, with a five-year survival rate of less than 35% of patients [60]. This circumstance pushes doctors to carefully observe the protocols for the study of biopsy–surgical material in pathological practice; in particular, it is imperative to reflect the state of the mucous membrane, which is the background for the development of a tumor.

## 3. Conclusions

The data presented in this article indicate the importance of implementing modern diagnostic capabilities for the earliest possible identification of patients with CAG. Unfortunately, in real practice, the diagnosis of CAG is significantly extended over time, probably due to a lack of awareness among doctors about the key principles of diagnosing the disease. Of course, many issues related to epidemiology, clinical features, diagnostic criteria, and tactics for managing patients with CAG require further study. Clinically, CAG often presents with anemia caused by vitamin B12 or iron deficiency. Polyneuropathy, various neuropsychiatric manifestations, and the presence of other autoimmune diseases in the anamnesis also alert clinicians to the conditions for possible CAG, especially in females.

It should be emphasized once again that atrophy of the mucous membrane in the stomach’s body, as a result of autoimmune inflammation, refers to precancerous changes and increases the risk of adenocarcinoma development; prolonged hypo- and achlorhydria are associated with an increase in the level of gastrin in the blood serum, a hormone that stimulates proliferation and hyperplasia of ECL cells, which in turn can contribute to the development of NETs of the gastric mucosa. Therefore, the adequate use of diagnostic tests, especially in the “risk” group, in relation to CAG (female persons with anemic syndrome and/or the presence of other autoimmune diseases) is necessary for the timely detection of this category of patient for subsequent follow-up and implementation of measures for the prevention of stomach cancer and NET.

## Figures and Tables

**Figure 1 diagnostics-11-02113-f001:**
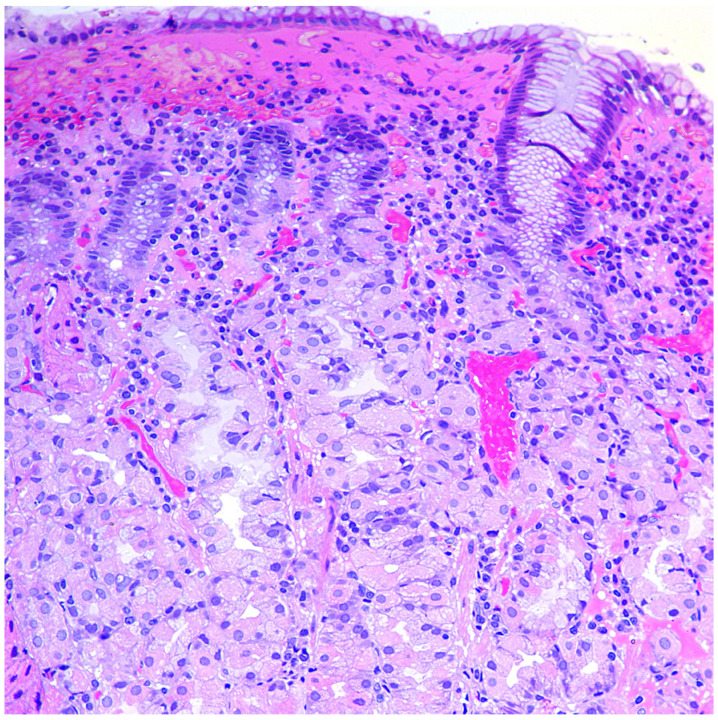
Autoimmune gastritis: initial stage. Biopsy from the gastric body. Lymphoplasmacytic inflammatory infiltration of the lamina propria with damage to the fundic glands. Hematoxylin and eosin stain. ×200 (from the personal archive of Professor S.I. Mozgovoi).

**Figure 2 diagnostics-11-02113-f002:**
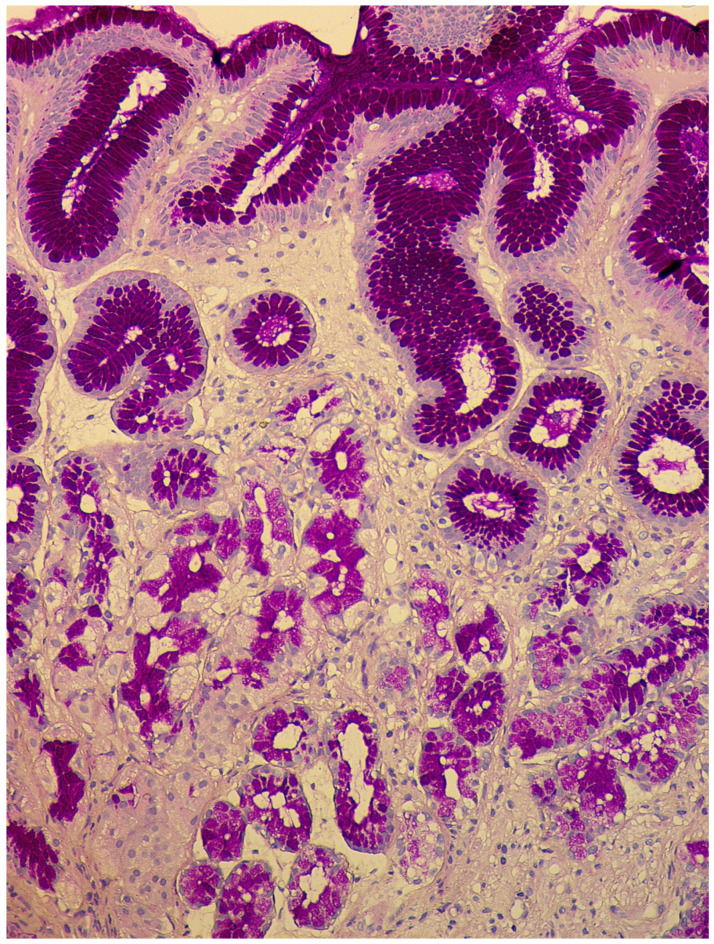
Autoimmune gastritis. Biopsy from the gastric body. Severe pseudopyloric metaplasia with the appearance of mucus-producing cells in place of the fundic glands (cytoplasm magenta staining). Periodic acid-Schiff stain. ×200 (from the personal archive of Professor S.I. Mozgovoi).

**Figure 3 diagnostics-11-02113-f003:**
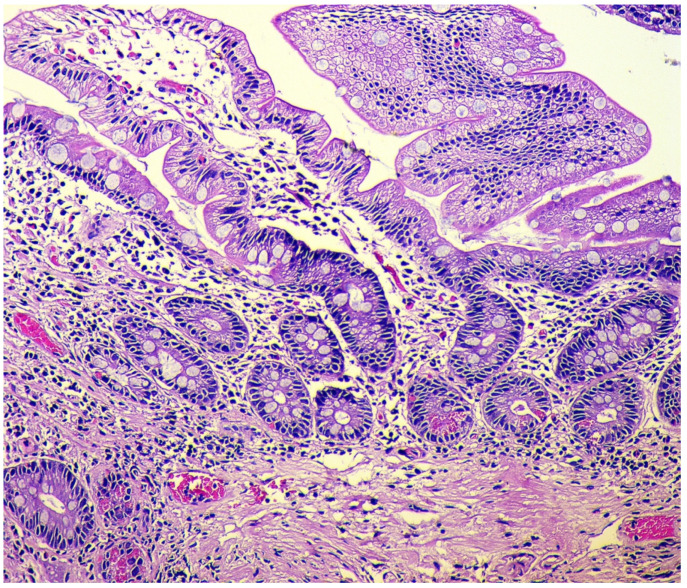
Autoimmune gastritis. Biopsy from the gastric body. Severe intestinal metaplasia (type I, complete), with intestinalization of gastric mucosa and villous transformation with absorptive and goblet cells; Paneth cells with acidophilic granular cytoplasm are presented at the base of the glands. Hematoxylin and eosin stain. ×200 (from the personal archive of Professor S.I. Mozgovoi).

**Figure 4 diagnostics-11-02113-f004:**
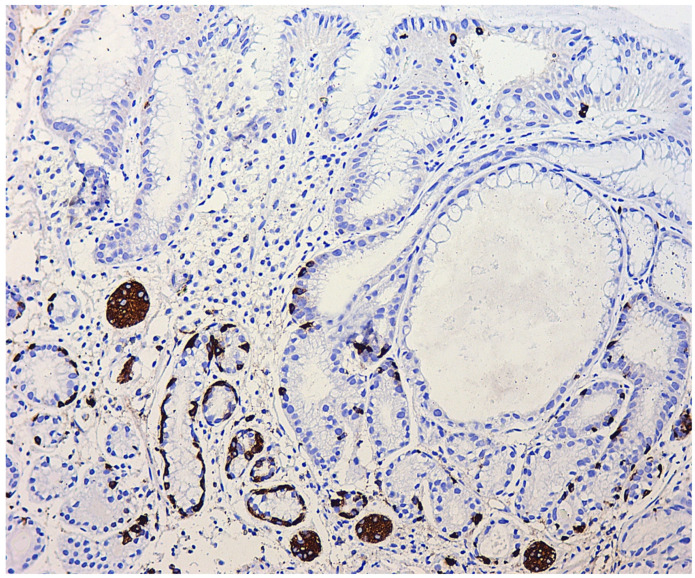
Autoimmune gastritis. Biopsy from the gastric body. Marked pseudopyloric metaplasia. Signs of linear and micronodular hyperplasia of neuroendocrine cells. Immunohistochemical reaction with antibodies to chromogranin A. ×200 (from the personal archive of Professor S.I. Mozgovoi).

**Figure 5 diagnostics-11-02113-f005:**
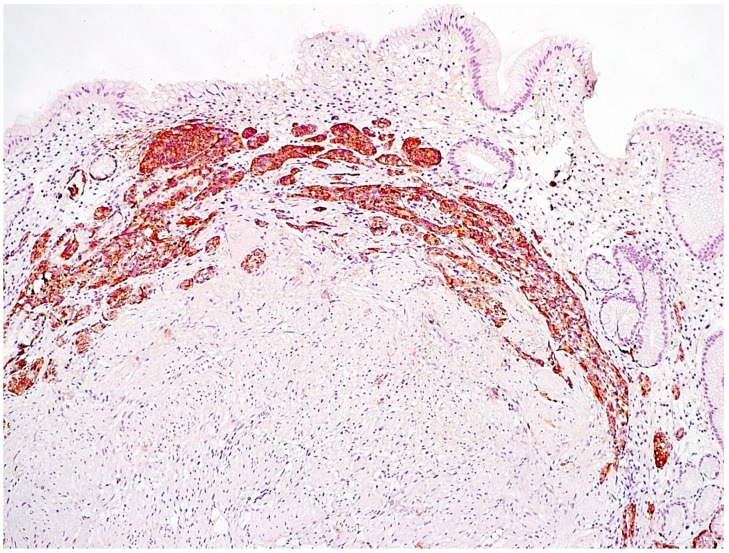
Autoimmune gastritis. Biopsy from the gastric body. Severe atrophy of the glands with pseudopyloric metaplasia. Adenomatous hyperplasia of neuroendocrine cells with the presence of cluster aggregates. Immunohistochemical reaction with antibodies to chromogranin A. ×100 (from the personal archive of Professor S.I. Mozgovoi).

**Table 1 diagnostics-11-02113-t001:** The main clinical manifestations of autoimmune gastritis (adapted from Massironi et al., 2019).

**Hematological Manifestations**
− Macrocytic anemia − Macrocytosis not associated with anemia − Iron deficiency anemia − Hemolytic anemia − Pancytopenia − Leukemoid reactions − Thrombosis associated with hyperhomocysteinemia
**Gastroenterological Manifestations**
− Symptoms of dyspepsia associated with motor disorders − Gastroesophageal reflux
**Psychoneurological Manifestations**
− Peripheral polyneuropathy − Myelopathy − Neuropathy of the optic and/or auditory nerve − Vegetative dysfunction − Depression − Cognitive impairment

**Table 2 diagnostics-11-02113-t002:** Micronutrient deficiency in autoimmune gastritis: prevalence, development mechanisms, clinical manifestations (adapted from Cavalcoli et al., 2017).

Micronutrient	Development Frequency	Development Mechanisms	Manifestations
Vitamin B12	37–69%	Decreased production of intrinsic Castle’s factor by parietal cells of the gastric body and decreased absorption of vitamin B12 in the ileum	Hematological, gastroenterological, neuropsychiatric disorders
Iron	52%	The presence of erosive lesions of the mucous membrane and possible latent blood loss, concomitant infection with H. pylori and bacteria competition for dietary iron, hypochlorhydria, and increased hepcidin synthesis against a background of a concomitant inflammatory process	Microcytic anemia
Vitamin C	Unknown	Breakdown of ascorbic acid in the stomach due to increased pH (hypo-, achlorhydria) and concomitant bacterial overgrowth in the small intestine	Decreased antioxidant defense, immunity, and protein synthesis
Calcium	Unknown	Dissolution, ionization, and absorption of calcium salts decrease under conditions of hypo-, achlorhydria	Osteopenia/Osteoporosis
Vitamin D	12.1%	Not determined	Secondary hyperparathyroidism, osteopenia/osteoporosis, decreased immunity, increased risk of autoimmune disease development

**Table 3 diagnostics-11-02113-t003:** Autoimmune diseases associated with autoimmune gastritis (adapted from Massironi et al., 2019).

Disease	Association Degree
Autoimmune thyroiditis (Hashimoto’s thyroiditis) and Graves’ disease	++++ (multiple cohort studies, two crossover studies, and two case–control studies)
Type 1 diabetes mellitus	+++ (one case–control study and several cohort studies)
Vitiligo	++ (one cohort study and several clinical observations)
Alopecia	++ (several cohort studies)
Celiac disease	++ (several cohort studies)
Myasthenia gravis	+ (single clinical observations)
Connective tissue disease	++ (one cohort study and several clinical observations)
Primary biliary cholangitis	++ (one cohort study and several clinical observations)
Primary sclerosing cholangitis	+ (single clinical observations)
Autoimmune hepatitis	+ (single clinical observations)
Addison’s disease	+ (single clinical observations)
Primary ovarian insufficiency	+ (single clinical observations)
Primary hypoparathyroidism	+ (single clinical observations)
Lambert-Eaton syndrome	+ (single clinical observations)
Oral lichen planus	+ (single clinical observations)

**Table 4 diagnostics-11-02113-t004:** Laboratory markers of CAG.

Indicant	Value	Comments
Antibodies to parietal cells	**↑**	They are found in 80–90% of patients with CAG, but they can also be present in 7.8–19.5% of healthy adults, as well as in people infected with HP or with other autoimmune diseases. They may not be determined in the later stages of the disease, as atrophy of the gastric mucosa progresses.
Antibodies to the intrinsic Castle’s factor	**↑**	Specificity—98.6%, sensitivity—30–60%. They correlate with atrophy of the gastric mucosa.
Pepsinogen I	**↓**	It is produced by the main cells of the stomach’s body. A decrease in the indicator shows the presence of atrophy in the mucous membrane of the stomach’s body. Serum level decreases in proportion to the severity of atrophy.
Pepsinogen II	Not changed	It is produced mainly in the stomach’s antrum and duodenum; therefore, it does not change with CAG.
Pepsinogen I/Pepsinogen II	**↓**	They decrease linearly with increasing severity of atrophy of the stomach’s body glands.
Gastrin-17	**↑**	It is produced by gastrin-producing cells of the stomach’s antrum. With a decrease in acid production against the background of the stomach’s body glands atrophy, it increases by the mechanism of a negative feedback loop.
Chromogranin A	**↑**	It correlates with the degree of ECL cell hyperplasia under conditions of hypo- and achlorhydria. Due to the low sensitivity and specificity, its widespread use in routine practice is limited.

## Data Availability

Not applicable.

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
