# Peer review of "Chronic Autoimmune Gastritis: Modern Diagnostic Principles"

_diagnostics, 2021, doi:10.3390/diagnostics11112113_

Round 1

Reviewer 1 Report

The significance of hyperhomocystinemia section in this paper is not understood well by the reader. Authors need to clarify on what they are trying to convey by including that sub section.

Chromagranin A being non specific, is not used to exclude the possible development of neuro endocrine tumor as stated by authors, this is only done histologically. Can authors share the reference here from where this information has been utilized?

The association between H pylori and hypertension is not referenced correctly. Where is this information utilized from.References 9-12 as quoted in this paper pertaining to this association do not mention this.

Chronic atrophic gastritis and intestinal metaplasia are very commonly encountered together in clinical practice, much more so then carcinoids or frank gastric carcinoma. For this reason, it would be beneficial for readers to know more about the types of intestinal metaplasia pathologically and a little about their significance with respect to risk of development of gastric cancer in setting of known CAG.

Author Response

1.The subsection “Hyperhomocysteinemia” was included by the authors in the section “Vitamin B12 (cobalamin) deficiency”, since the homocysteine increase in serum can be observed in early stages, sometimes before the clinical manifestations of the vitamin B12 deficiency. Since hyperhomocysteinemia is a proven risk factor for cardiovascular diseases, the authors wanted to emphasize that this fact should be taken into account when managing patients with CAG.

2.In accordance with the reviewer’s recommendations, the “Laboratory Methods” section was supplemented with references to studies that assessed the possibility of using chromogranin A as a possible marker of NET. It was pointed out that chromogranin A is nonspecific and can only be considered as an auxiliary marker.

3.Regarding the link between H. pylori and arterial hypertension, the authors apologize and report a translation error. This article described autoimmune gastritis, but not arterial hypertension. In the Russian-language version of the article, the abbreviation “АH” was used, which was mistakenly translated as arterial hypertension.

4.In the section “Morphological examination”, we added information on the intestinal metaplasia types and included new references to modern studies on this topic.

Reviewer 2 Report

In this narrative review, the authors summarize data about chronic autoimmune gastritis (CAG). The authors focused their attention on risk factors, clinical manifestations concerning gastrointestinal symptoms and haematological alterations and comorbidities. The second part of the narrative review focuses on diagnostic approaches, regarding the laboratory, endoscopic and histological methods.

Comments:

Chronic autoimmune gastritis is an underestimated condition that could have not only gastrointestinal implications but even haematological and neurological symptoms.

This is an interesting review because it summarizes available data to diagnose this condition. However, some comments are necessary:

  1. In Diagnostic approaches, concerning the use of Chromogranin A, the authors asserted that “To exclude the possible development of a neuroendocrine tumor of the gastric mucosa against the background of CAG, it is advisable to determine Chromogranin A…”. However, due to the low sensitivity and specificity of this marker, it is impossible to exclude the possible development of neuroendocrine tumours only using this serological marker. The authors should soften this sentence.
  2. The endoscopic examination section lacks the modern diagnostic techniques useful to diagnose chronic autoimmune gastritis and the use of electronic chromoendoscopy to target biopsies and to recognize neuroendocrine tumours. The authors should emphasize how the image enhanced endoscopy facilitates the diagnosis of chronic autoimmune gastritis and in particular in the intestinal metaplasia stadium, permitting a correct staging in the corpus mucosa.
  3. It would be advisable to cite even MAPS II guidelines that in contrast to the first version include autoimmune gastritis as a risk factor for follow-up
  4. In the morphological examination section, the latest articles about pseudopyloric metaplasia and the risk of gastric cancer should be cited.
  5. Furthermore, the authors asserted that the OLGIM system is not so widespread in clinical practice. However, it is impossible to support with data this sentence because of the lack of studies that analyze data about the real-life histopathological evaluation. For sure, the OLGIM classification is more reliable as the agreement on the diagnosis of atrophy is low, in contrast with the diagnosis of intestinal metaplasia that is much more reliable. The authors should emphasize this difference in the agreement concerning the use of these classifications. Indeed, due to the recognition of intestinal metaplasia by virtual chromoendoscopy, staging systems for endoscopic assessment of intestinal metaplasia were related to the OLGIM classification.
  6. Concerning neuroendocrine tumours, the authors should describe the diagnosis and management of these tumours.

Author Response

1.In accordance with the recommendations of the reviewer, in the “Laboratory Methods” section, the phrase “To exclude the possible development of a neuroendocrine tumor (NET) of the gastric mucosa against the background of CAG, it is advisable to determine chromogranin A ...” was softened with the note that Chromogranin A “… can be considered as an auxiliary marker for assessing the risk of NET formation. The sensitivity and specificity of chromogranin A as a possible marker of NET is low, therefore its determination result should be assessed taking into account the clinical picture and the histological examination data.” References to studies that assessed the possibility of using chromogranin A as a possible marker of NET were added.

2.The section “Endoscopic examination” was supplemented with modern diagnostic methods that can be used to diagnose CAG, including the possibility of using electronic chromoendoscopy for making a targeted biopsy and excluding NET.

3.The MAPS II recommendations were cited, including a statement on the observation frequency for patients with CAG.

4.The section “Morphological examination” was supplemented with references to modern publications.

5.With regard to the use of morphological classifications OLGA and OLGIM, we indicated that “The modified OLGIM system (Operative Link on Gastric Intestinal Metaplasia Assessment), being more reproducible, offers only an assessment of the intestinal metaplasia presence as the main indicator of chronic gastritis stage. Although the question of comparing these classifications from the sensitivity point remains controversial, since the use of OLGIM, which is more reproducible in practical application, may underestimate the true severity of absolute (non-metaplastic) atrophy. “

6.Regarding the reviewer’s recommendations for an expansion of the section on the Diagnosis and Treatment of NET, ​​the authors agree that this topic is relevant and complex, but they did not set this as the goal of this review. The authors considered only the initial manifestations of neuroendocrine hyperplasia as an integral part of the morphological picture in patients with autoimmune atrophic gastritis.

Round 2

Reviewer 1 Report

The section on laboratory methods is referring to "hypertension" multiple times again. Is it again an error and it should be CAG instead of hypertension ?

I  still don't see the description of different types of intestinal metaplasia( focal, extensive; complete, incomplete etc) in the "morphological examination" section 

Author Response

Review 1

The authors apologize and report this is a similar translation error. The indicated error has been fixed.

In accordance with the recommendations of the reviewer, in the section “Morphological examination” we added information of different types of intestinal metaplasia.

Round 3

Reviewer 1 Report

No further comments